# Pt Nanoparticles Supported on Ultrathin Ni(OH)$_2$ Nanosheets for Highly Efficient Reduction of 4-Nitrophenol

Jia-Lin Cui [1], Zhong-Liang Liu [1], Hui-Hui Li [1,*] and Chun-Zhong Li [1,2,*]

[1] Key Laboratory for Ultrafine Materials of Ministry of Education, School of Chemical Engineering, East China University of Science and Technology, Shanghai 200237, China; y30200107@mail.ecust.edu.cn (J.-L.C.); y11200009@mail.ecust.edu.cn (Z.-L.L.)

[2] Shanghai Engineering Research Center of Hierarchical Nanomaterials, School of Materials Science and Engineering, East China University of Science and Technology, Shanghai 200237, China

* Correspondence: huihuili@ecust.edu.cn (H.-H.L.); czli@ecust.edu.cn (C.-Z.L.)

**Abstract:** The synthesis of highly efficient heterogeneous catalysts with uniformly dispersed noble metal particles and a suitable size is crucial for various industrial applications. However, the high cost and rarity of noble metals limit their economic efficiency, making it essential to improve the catalytic performance with lower noble metal loading. Herein, a two-step method was developed for the synthesis of uniformly dispersed ~3 nm Pt nanoparticles (NPs), strongly anchored on Ni(OH)$_2$ nanosheets (NSs), which was proven by adequate structural characterizations. XPS analysis demonstrated that Ni(OH)$_2$ NSs with abundant oxygen vacancies provided sufficient anchor sites for Pt NPs and prevented their agglomeration. The catalytic performance of Pt$_n$/Ni(OH)$_2$ (n (represents the addition amount of Pt precursors during the synthesis, μmol) = 5, 10, 15, and 20) NSs with controllable Pt loading were evaluated via the reduction of 4-nitrophenol to 4-aminophenol as a model reaction. The Pt$_{10}$/Ni(OH)$_2$ NSs exhibited the best activity and stability, with a reaction rate constant of 0.02358 s$^{-1}$ and negligible deterioration in ten reaction cycles. This novel synthetic method shows potentials for the synthesis of highly efficient noble-metal-supported catalysts for heterogeneous catalysis.

**Keywords:** Pt/Ni(OH)$_2$ nanosheets; oxygen vacancy; heterogeneous catalysis; reduction of 4-nitrophenol





## 1. Introduction

Heterogeneous catalysts are crucial in creating a sustainable world by reducing energy consumption, carbon emissions, and environmental pollution during manufacturing. Among them, noble-metal-based catalysts have been extensively studied in academia and utilized across various industrial applications (e.g., the petrochemical industry, electrochemistry, energy conversion, and pollutant elimination) due to their unique electronic structure and surface chemistry, which enable them to activate various molecules by donating or accepting electrons [1–4]. For instance, platinum (Pt) is a promising selection in the catalytic field [5]. However, the costly and rare noble metals limit the economic efficiency of catalytic processes, which leads to a strong demand for the enhancement of catalytic performance and a reduction in noble metal loading.

Considerable efforts have been spent on both enhancing the catalytic performance and maximizing the atomic utilization of noble metals. Structure engineering is an effective strategy to expand the specific area and expose more active sites. The catalytic activity and selectivity of noble metal catalysts could be modulated to achieve the desired target by constructing 1D structures (nanowires, nanorods, and nanotubes) [6], 2D structures (nanosheets and nanoribbons) [7], and 3D structures (core–shell structures) [8]. Additionally, the particle size effect is another important factor which can heavily influence the catalytic performance of noble metal catalysts through the exposure of more active sites

and modulation of the electronic structure (orbital overlapping) and geometric structure (edge, corner, facets, etc.) [9]. Furthermore, the interaction between the noble metal and the support materials would significantly alter the electronic structures of metal active sites, adjusting the adsorption energetics of reaction intermediates and affecting activity and selectivity [10,11]. The metal–support interaction could also help to stabilize the metal catalysts and suppress their aggregation during the synthesis and catalysis process [12]. Based on the above discussion, a highly active heterogeneous catalyst should have uniformly dispersed noble metal particles of suitable size, stably anchored on the nanostructured materials. However, how to achieve the accurate synthesis of such structures with a high catalytic performance remains a challenge.

Herein, we explored a simple two-step method that enables the in-situ deposition of uniformly dispersed, loading-controllable ~3 nm Pt nanoparticles (NPs) onto $Ni(OH)_2$ NSs ($Pt/Ni(OH)_2$ NSs). The $Ni(OH)_2$ NSs with flat planes and abundant vacancies could effectively anchor and disperse the Pt NPs without agglomeration at room temperature [13,14]. The Pt loading could be modulated in the second step, further adjusting the catalytic performance. The reduction of 4-nitrophenol (4-NP) to 4-Aminophenol (4-AP) as a model heterogeneous catalytic reaction was employed to evaluate the catalytic performance of the synthesized $Pt_n/Ni(OH)_2$ NSs ($n$ = 5, 10, 15, 20) [15,16]. Among all obtained $Pt_n/Ni(OH)_2$ NSs samples, $Pt10/Ni(OH)_2$ NSs exhibited the best activity and stability in catalyzing 4-NP's reduction to 4-AP, with a kinetic rate constant of 0.02358 $s^{-1}$ and negligible deterioration in 10 reaction cycles. Our simple synthetic method shows great potential in synthesizing noble-metal-supported catalysts with high and uniform dispersion for heterogeneous catalysis.

## 2. Results and Discussion

### 2.1. Synthesis and Characterizations

The $Pt_n/Ni(OH)_2$ NSs were synthesized using a two-step method, which involved the pre-synthesis of $Ni(OH)_2$ NSs and the subsequent in-situ anchoring of Pt NPs on the $Ni(OH)_2$ NSs. In the first step, the $Ni(OH)_2$ NSs were synthesized via a hydrothermal method using a round-bottomed flask as a reactor. In the second step, the room-temperature chemical reduction in an aqueous solution effectively limited the agglomeration of Pt NPs. Such a stable anchoring effect played an important role in the uniform dispersion of the Pt NPs on $Ni(OH)_2$ NSs. The Pt loading was controlled by adjusting the addition amount of Pt precursor during the synthesis process. Four $Pt_n/Ni(OH)_2$ NSs with different Pt loadings were synthesized, and the mass percentage of Pt in each sample was obtained via ICP-OES, as shown in Table S1.

The morphology and structure of the as-prepared $Ni(OH)_2$ NSs and $Pt_{10}/Ni(OH)_2$ NSs (selected as an example) were first investigated via electron microscopies. Scanning electron microscopy (SEM) images showed that the as-prepared $Ni(OH)_2$ NSs had a flaky morphology with an ultrathin thickness (Figure 1a,b). Transmission electron microscopy (TEM) and high-resolution transmission electron microscopy (HR-TEM) were employed to observe the structure of $Pt_{10}/Ni(OH)_2$ NSs and the size of anchored Pt NPs. Figure 1d showed that the Pt NPs were uniformly dispersed onto the ultrathin and flat $Ni(OH)_2$ NSs without agglomeration, successfully preserving the flaky morphology and forming the $Pt_n/Ni(OH)_2$ NSs. Figure 1e and the insert showed that the size of the anchored Pt NPs was measured as 3.0 ± 0.66 nm, indicating the advantage of our room-temperature synthetic method at anchoring Pt NPs on nanostructured supports. The corresponding electron diffraction (SAED) patterns of a selected area of $Ni(OH)_2$ NSs and $Pt_{10}/Ni(OH)_2$ NSs both showed two bright rings corresponding to (100) and (110) facets of $Ni(OH)_2$, respectively. To note, the bright spots were clearer in the $Pt_{10}/Ni(OH)_2$ NSs than that in the $Ni(OH)_2$ NSs, possibly due to the enhanced crystallinity resulting from anchored Pt NPs (Figure 1c,f).

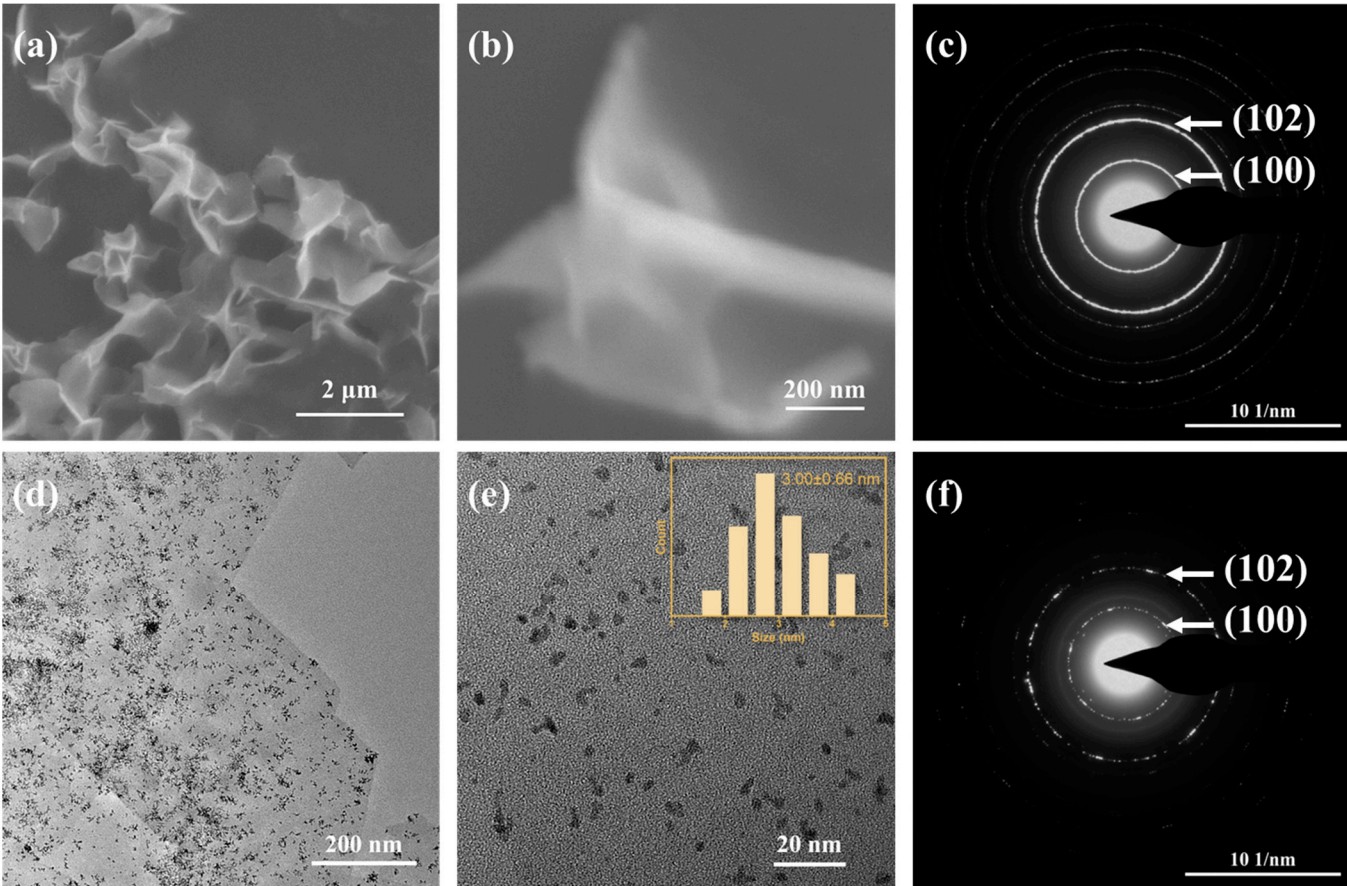

**Figure 1.** (**a**,**b**) SEM images of Ni(OH)$_2$ NSs. (**c**) SAED pattern of Ni(OH)$_2$ NSs. (**d**) TEM images of Pt$_{10}$/Ni(OH)$_2$ NSs and (**e**) HR-TEM images of Pt$_{10}$/Ni(OH)$_2$ NSs; the insert bar chart showed the average size of Pt NPs. (**f**) SAED pattern of Pt$_{10}$/Ni(OH)$_2$ NSs.

The crystal structures of the as-prepared samples were further analyzed via X-ray diffraction (XRD). The XRD patterns of all samples showed characteristic peaks of Ni(OH)$_2$ (PDF 14-0117), in which two intense and sharp peaks were assigned to (100) and (110) facets of Ni(OH)$_2$, concordant with the SAED results (Figure 2a). As the Pt loading increased, the characteristic peaks of metallic Pt (PDF 04-0802) gradually emerged at $2\theta \approx 39.8°$ and 46.2°, confirming the controllable Pt loading by our synthetic method. The emergence of metallic Pt peaks also indicated that the Pt amount was excessive, leading to the aggregation and the increased size of the Pt NPs. Additionally, the characteristic peak of the basal plane (001) at $2\theta \approx 19.2°$ slightly shifted to a low angle in Pt$_n$/Ni(OH)$_2$ NSs, resulting from the Pt NPs loading onto the surface of nanosheets instead of intercalating into the interlayer due to the narrow basal spacing (~4.6 Å). This was also verified by the magnified XRD patterns (Figure S1) which showed that no peaks were observed in the range of $2\theta = 5–18°$.

The lamellar structure of Pt$_n$/Ni(OH)$_2$ NSs was also verified via Fourier-transform infrared spectroscopy (FT-IR), as shown in Figure 2b. The peak at 3641 cm$^{-1}$ in every sample was identified as the OH$^-$ stretching vibration peak in Ni(OH)$_2$. Moreover, all samples exhibited strong peaks at 3400 cm$^{-1}$ and 1620 cm$^{-1}$, which were assigned to the stretching vibration peak and bending vibration peak of H$_2$O, respectively. This result suggested that the Pt$_n$/Ni(OH)$_2$ NSs had a lamellar structure that easily adsorbs water, further affirming the successful synthesis of Pt NPs supported by nanosheets.

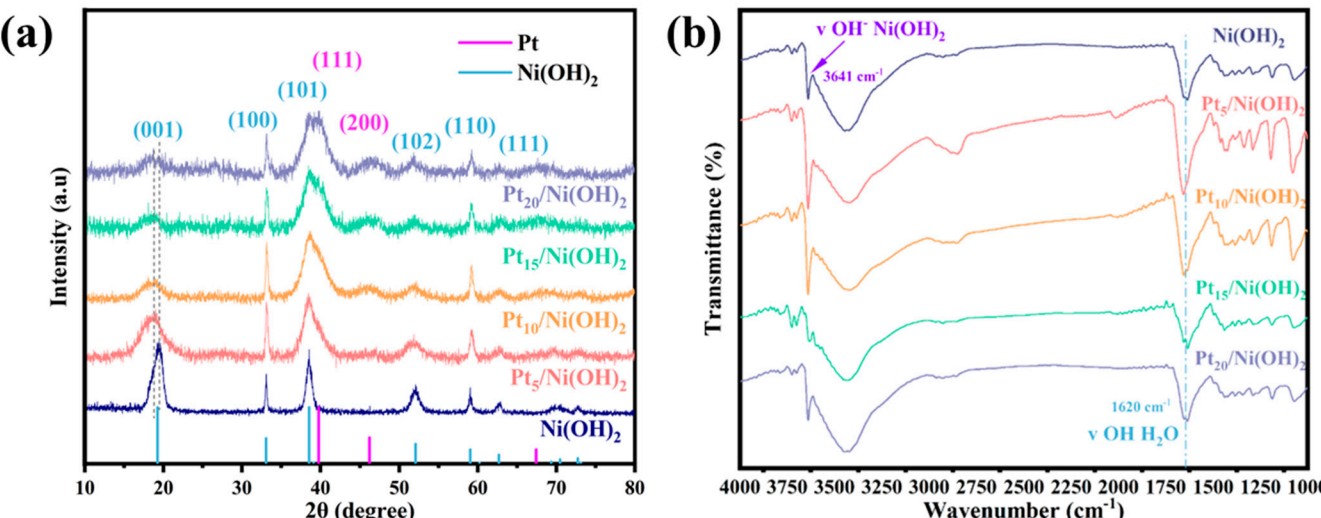

**Figure 2.** (**a**) XRD patterns of as-prepared Ni(OH)$_2$ NSs and Pt$_n$/Ni(OH)$_2$ NSs with different Pt loading. The standard Ni(OH)$_2$ (PDF 14−0117) and Pt (PDF 04−0802) are also plotted as references. (**b**) FT-IR spectra of Ni(OH)$_2$ NSs and Pt$_n$/Ni(OH)$_2$ NSs with different Pt loadings.

Considering that the nanostructured support was beneficial to expose more anchor sites for Pt NPs, the specific surface areas of all samples were evaluated via nitrogen adsorption–desorption isotherms and calculated using the Brunauer–Emmett–Teller (BET) method. The Ni(OH)$_2$ NSs had a large specific surface area of 57.67 m$^2$/g, and the specific surface area first increased and then decreased along with the increase in Pt loading (Table S2). This trend was in line with the XRD results, which indicated that the excessive addition of the Pt precursor during the synthesis process might cause the enlargement of the size of Pt NPs or the aggregation of Pt NPs, leading to the decrease in the specific surface area. This observation was also ascertained via the TEM images of Pt$_{10}$/Ni(OH)$_2$ and Pt$_{15}$/Ni(OH)$_2$ NSs, which showed the aggregation of Pt NPs, and that the size of Pt NPs increased to 3.38 ± 0.67 and 6.28 ± 2.04 nm, respectively (Figure S2). Notably, Pt$_5$/Ni(OH)$_2$ has a smaller particle size (2.52 ± 0.52 nm) but a lower specific surface area than Pt$_{10}$/Ni(OH)$_2$, which can be attributed to a too-low Pt loading and consequently fewer active sites. By comparing with the other three samples, the Pt$_{10}$/Ni(OH)$_2$ NSs possessed the largest specific surface areas of 104.47 m$^2$/g, hinting that Pt$_{10}$/Ni(OH)$_2$ NSs exhibit more uniformly dispersed Pt NPs on the nanosheet and expose more active sites.

### 2.2. Mechanism of Pt Anchoring on the Ni(OH)$_2$ NSs

To investigate how Pt NPs in-situ anchored on the Ni(OH)$_2$ NSs, we employed X-ray photoelectron spectroscopy (XPS) to study their chemical state changes after anchoring different amounts of Pt NPs. We first analyzed the chemical state of Ni species from the Ni 2p XPS spectrum. Both the Ni 2p XPS spectrum of Ni(OH)$_2$ NSs and Pt$_{10}$/Ni(OH)$_2$ NSs showed two prominent peaks of Ni 2p$_{3/2}$ at 855.6 eV and Ni 2p$_{1/2}$ at 873.2 eV, which were attributed to the Ni$^{2+}$ valence state of Ni(OH)$_2$ [17], implying that Pt anchoring does not interact with Ni species (Figure 3a,d).

We then turned to analyzing the changes in the oxygen species. The O 1s XPS spectra could be fitted to three peaks which could be ascribed to lattice oxygen, oxygen vacancy, and oxygen from adsorbed water molecules, respectively [18]. Figure 3b,e showed that the oxygen vacancy density decreased after the Pt NP anchoring, which triggered a hypothesis that oxygen vacancies on the surface of Ni(OH)$_2$ NSs provide the anchor sites for Pt NPs. We further calculated the oxygen vacancy density of all four samples, and the results were in agreement with our hypothesis. That is, the oxygen vacancy density decreased along with the increase in Pt loading, meaning the Pt NPs replenish the oxygen vacancies on the surface of Ni(OH)$_2$ NSs, as shown in Figure 3f. Notably, the analysis of oxygen

vacancy density concurred with the BET specific surface area results. That is, the oxygen vacancy density was almost the same in $Pt_{15}/Ni(OH)_2$ NSs and $Pt_{20}/Ni(OH)_2$ NSs, which suggested the Pt loading for $Pt_{15}/Ni(OH)_2$ NSs approached the limit and excessive Pt might be adverse for the synthesized $Pt_n/Ni(OH)_2$ NSs.

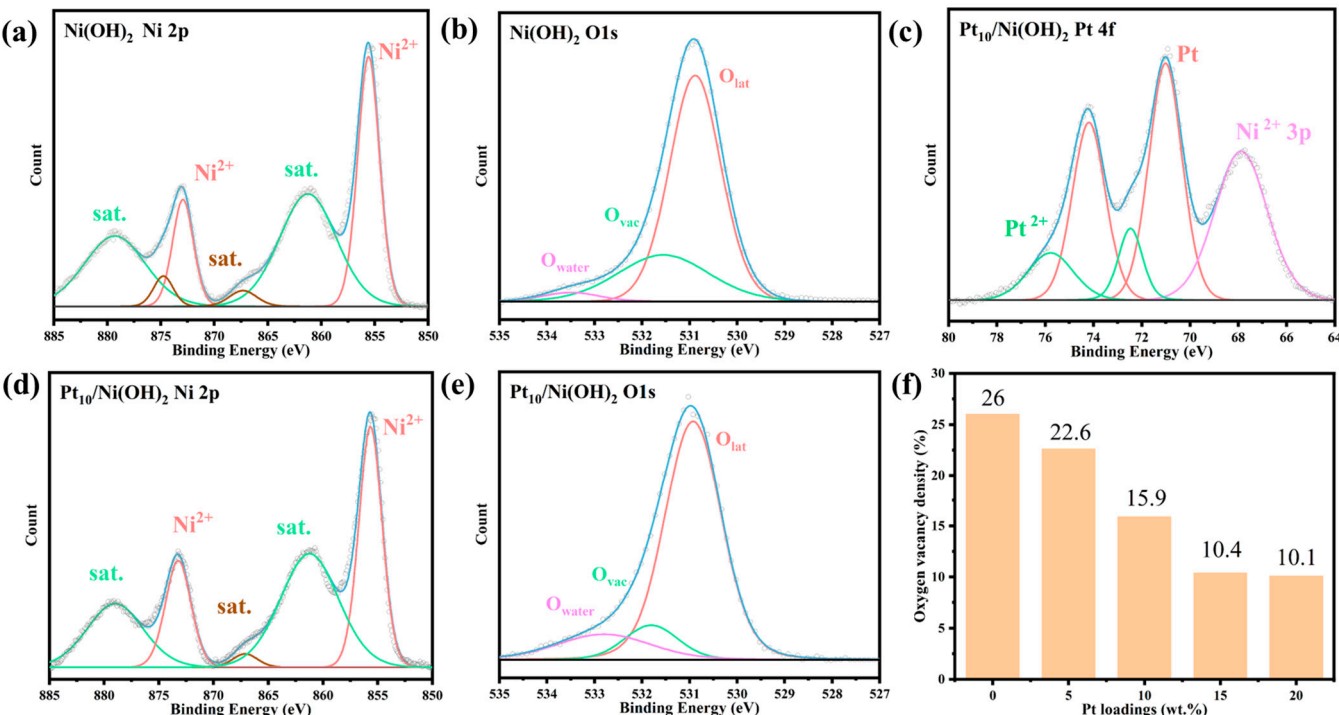

**Figure 3.** (**a**) Ni 2p and (**b**) O1s XPS spectra of Ni(OH)2 NSs, respectively. (**c**) Pt 4f, (**d**) Ni 2p, and (**e**) O 1s XPS spectra of $Pt_{10}/Ni(OH)_2$ NSs, respectively. (**f**) Oxygen vacancy density of $Pt_n/Ni(OH)_2$ surface with different Pt loading amounts calculated from O 1s XPS spectra. All XPS spectrum were baseline subtracted.

The Pt 4f XPS spectra of $Pt_{10}/Ni(OH)_2$ NSs further affirmed the above hypothesis, which showed two Pt valence states of $Pt^0$ and $Pt^{2+}$ at 71.01 eV and 72.46 eV (Figure 3c), respectively. The $Pt^{2+}$ valence state was associated with $Pt(OH)_2$ [19], indicating that Pt anchoring interacts with the hydroxyl at the interface between Pt NPs and $Ni(OH)_2$ NSs, further illustrating that Pt anchoring is closely related with oxygen vacancy. To note, the Pt 4f XPS spectra overlapped with the Ni 3p XPS spectra of $Ni(OH)_2$, which was resolved and fitted in Figure 3c.

### 2.3. Evaluation of Heterogeneous Catalytic Performance

The heterogeneous catalytic performance of various $Pt_n/Ni(OH)_2$ NSs and a commercial Pt/C catalyst were evaluated via the reduction of 4-NP to 4-AP in the presence of $NaBH_4$. The reduction of 4-NP to 4-AP is a widely used model reaction to evaluate the catalytic performance of various nanostructured materials, especially noble metal NPs [20]. In addition, the reduction reaction rate and conversion of 4-NP under ambient conditions can be fast and easily monitored using UV-vis spectroscopy, which is beneficial to accelerate the development of heterogeneous catalysts. Therefore, this reaction has been employed as an effective model reaction to evaluate the activity and stability of various heterogeneous catalysts.

Figure 4a shows that the absorbance peak of 4-NP at 317 nm shifted to 400 nm, which is assigned to the 4-nitrophenolate ion after adding $NaBH_4$. The catalytic performance of pure $Ni(OH)_2$ NSs was first measured as a blank experiment. As shown in Figure 4b, the absorbance of the 4-nitrophenolate ion was nearly unchanged after 15 min of reaction, which

means the Ni(OH)$_2$ NSs did not have catalytic activity for the reduction of 4-NP to 4-AP and the NaBH$_4$ did not trigger this reaction either. The catalytic activity of Pt$_n$/Ni(OH)$_2$ NSs with different Pt loadings was then evaluated, as shown in Figure 4d–g. The absorbance peak of the 4-nitrophenolate ion decreased along with the reaction time, demonstrating that the obtained excellent catalytic activity is attributed to the supported Pt NPs. After 10 min of reaction, the absorbance curve flattened for all the Pt$_n$/Ni(OH)$_2$ NSs, suggesting an almost 100% conversion of 4-NP.

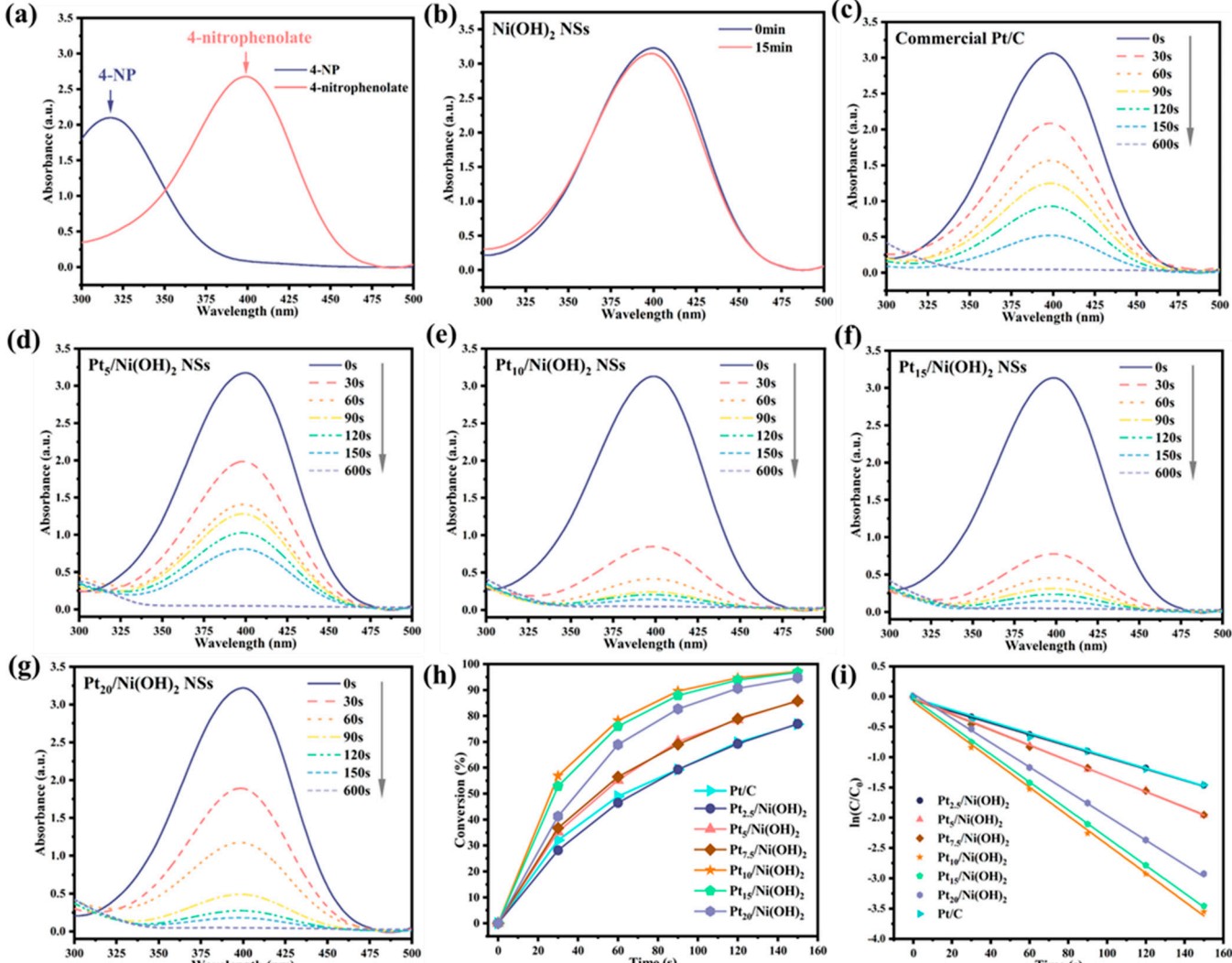

**Figure 4.** UV–vis absorption spectra of (**a**) 4-NP before and after adding the NaBH$_4$ solution without the catalyst. (**b**) 4-NP with added NaBH$_4$ and Ni(OH)$_2$ NSs. (**c**) 4-NP with added NaBH$_4$ and commercial Pt/C. (**d**–**g**) 4-NP with added NaBH$_4$ and Pt$_n$/Ni(OH)$_2$ NSs (*n* = 5, 10, 15, 20). (**h**) Time-dependent conversion of 4-NP catalyzed by various samples. (**i**) Relationship of ln(C/C$_0$) and reaction time for the reduction of 4-NP catalyzed by various samples.

Further analysis of the relationship between the absorbance curves and the reaction time indicated that the reaction rate of Pt$_n$/Ni(OH)$_2$ NSs varies with Pt loading. Specifically, the reaction rate first accelerated and then decelerated along with the increased Pt loading. The Pt$_{10}$/Ni(OH)$_2$ NSs and Pt$_{15}$/Ni(OH)$_2$ NSs exhibited superior activity among all samples, as shown in Figure 4e,f. The concentration of the 4-nitrophenolate ion dramatically decreased after only 30 s and almost converted completely in 150 s. In comparison with the commercial Pt/C catalyst, all Pt$_n$/Ni(OH)$_2$ NSs showed better catalytic activities, which indicated that the nanostructured support with a high specific surface area and the

uniform, dispersed, small-sized Pt NPs are the two main factors to improve the activity of the reduction of 4-NP (Figure 4h). Furthermore, the quantitative analysis of reaction kinetics was performed by fitting the reaction rate curve:

$$-\ln(C/C_0) = kt, \tag{1}$$

where C and $C_0$ refer to the absorbance of the 4-nitrophenolate ion in the current and initial solutions, respectively. $k$ and $t$ refer to the reaction rate constant ($s^{-1}$) and the reaction time (s), respectively.

Figure 4i illustrates that the linear fitting of the reaction rate curve was valid for all samples (including $Pt_{2.5}/Ni(OH)_2$ NSs and $Pt_{7.5}/Ni(OH)_2$ NSs), suggesting a first-order reaction for the reduction of 4-NP catalyzed by heterogeneous catalysts. The reaction rate constant $k$ can serve as a descriptor to compare the activity between different catalysts, as shown in Table S3. The commercial Pt/C catalyst exhibited a relatively low reaction rate constant of 0.00967 $s^{-1}$ and a conversion of 98.4% after 10 min of reaction, which demonstrated that our synthesized $Pt_n/Ni(OH)_2$ NSs with nanostructured support surpassed the widely used Pt/C catalyst for heterogeneous catalysis. Additionally, a volcano relationship could be identified between the reaction rate constant $k$ and the Pt loading, which showed an enhancement of activity from $Pt_5/Ni(OH)_2$ NSs (0.01289 $s^{-1}$) to $Pt_{10}/Ni(OH)_2$ NSs (0.02358 $s^{-1}$), while a decline in activity was found when Pt loading further increased (0.002294 $s^{-1}$ for $Pt_{15}/Ni(OH)_2$ NSs and 0.01973 $s^{-1}$ for $Pt_{20}/Ni(OH)_2$ NSs). This relationship was in agreement with our characterization results, which showed that the size of Pt NPs increased with the Pt loading, which is the key factor to influence the activity of $Pt_n/Ni(OH)_2$, resulting in the decrease in activity for $Pt_{15}/Ni(OH)_2$ NSs and $Pt_{20}/Ni(OH)_2$ NSs. Although $Pt_5/Ni(OH)_2$ NSs had the smallest size of the Pt NPs, its activity was inferior to the other samples. This can be attributed to the low Pt loading and the resulting limited number of active sites. Therefore, the trend in activity was consistent with the BET specific surface area, which initially increased and then decreased with Pt loading. Consequently, $Pt_{10}/Ni(OH)_2$ NSs possessed the largest specific surface area and the highest activity among all the samples.

The mass-specific activity was crucial for noble-metal-based heterogeneous catalysts due to the high cost of noble metals. Thus, the mass-normalized reaction rate constant ($k_m$) was calculated as shown in Table S3, which pointed out that the $Pt_5/Ni(OH)_2$ NSs and $Pt_{10}/Ni(OH)_2$ NSs have a similarly high $k_m$ of 224.56 $s^{-1} \cdot g^{-1}$ and 214.27 $s^{-1} \cdot g^{-1}$, respectively. However, the mass-specific activity of $Pt_{15}/Ni(OH)_2$ NSs and $Pt_{20}/Ni(OH)_2$ NSs drastically degraded to 155.53 $s^{-1} \cdot g^{-1}$ and 110.13 $s^{-1} \cdot g^{-1}$, respectively, when the added amount of Pt precursor exceeded 10 μmol. This phenomenon indicated that the Pt NPs could uniformly be dispersed on the $Ni(OH)_2$ NSs and achieved high mass-specific activity if the amount of precursor is below 10 μmol. However, the Pt NPs will aggregate if the Pt precursor is excessive, which results in the enlarged size of NPs and the low utilization of noble metal atoms (Figure S2). This speculation was also concordant with the XRD results, TEM images, XPS analysis of oxygen vacancy, and the trend of the BET specific surface area. Notably, all of the synthesized $Pt_n/Ni(OH)_2$ NSs had a higher $k_m$ than the commercial Pt/C catalyst, demonstrating the advantages of the nanosheet structure for the exposure of noble metal active sites.

The stability of catalysts was evaluated via a continuous reduction reaction for 10 cycles. Figure 5 shows the repeatability of the conversion of 4-NP for all samples in 10 cycles, which revealed the excellent reactive stability for $Pt_n/Ni(OH)_2$ NSs, except for the $Pt_{20}/Ni(OH)_2$ NSs. The relatively poor stability of $Pt_{20}/Ni(OH)_2$ NSs could be responsible for the weakening binding strength between Pt NPs and the $Ni(OH)_2$ NS support due to the aggregation of Pt NPs. Further detailed statistical analysis of the conversion during 10 cycles confirmed the superior cyclic stability of $Pt_n/Ni(OH)_2$ NSs, except for the $Pt_{20}/Ni(OH)_2$ NSs (Table S4). In particular, the most active $Pt_{10}/Ni(OH)_2$ NSs also exhibited the best stability, with a variance of 0.12 and a standard deviation of 0.46, surpassing the other three $Pt_n/Ni(OH)_2$ NSs.

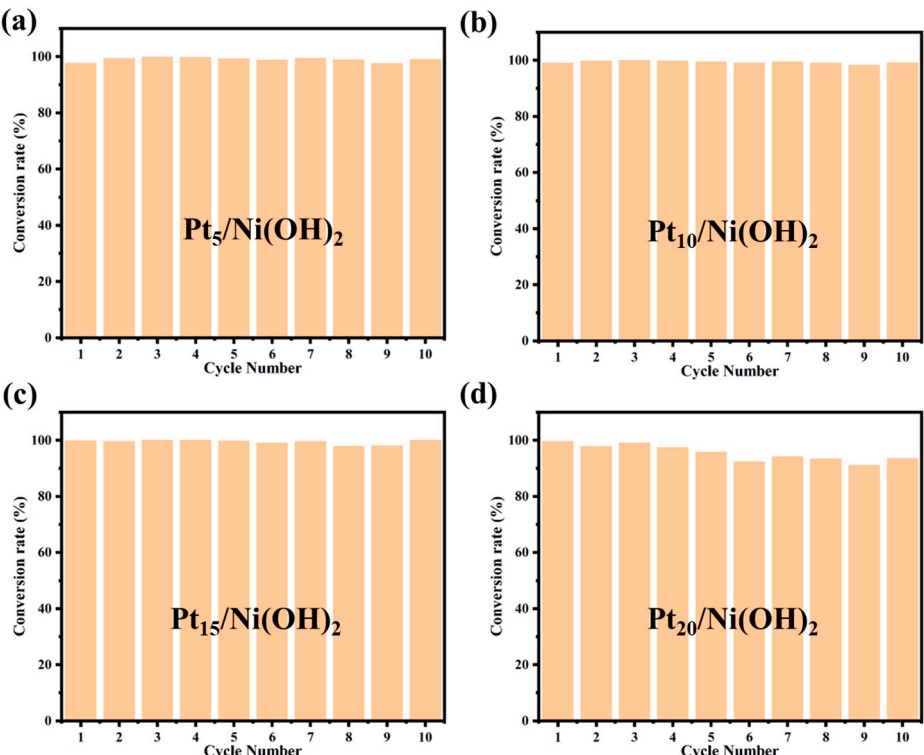

**Figure 5.** Conversion of 4-NP catalyzed by (**a**) $Pt_5/Ni(OH)_2$ NSs, (**b**) $Pt_{10}/Ni(OH)_2$ NSs, (**c**) $Pt_{15}/Ni(OH)_2$ NSs, and (**d**) $Pt_{20}/Ni(OH)_2$ NSs in 10 reaction cycles.

### 3. Experimental Section

*3.1. Chemicals*

Nickel(II) chloride hexahydrate ($NiCl_2 \cdot 6H_2O$, AR) was purchased from Adams-beta (Shanghai, China), while hexamethylenetetramine (HMT, AR) and sodium borohydride ($NaBH_4$, AR) were purchased from Sinopharm (Beijing, China). Potassium tetrachloroplatinate(II) ($K_2PtCl_4$, AR) was purchased from Aladdin (Shanghai, China), sodium formate (HCOONa, AR) was purchased from Macklin (Shanghai, China), and polyvinyl pyrrolidone (PVP, MW 55000) was purchased from Shanghai yuanye Bio-Technology Co., Ltd. (Shanghai, China). A commercial Pt/C (20 wt.%) catalyst was purchased from Johnson Matthey (London, UK). All reagents were used without any further purification.

*3.2. Synthesis of Ni(OH)₂ Nanosheets*

A total of 0.78 g of $NiCl_2 \cdot 6H_2O$ and 2.33 g of HMT were firstly dissolved in 400 mL of deionized water in a round-bottomed flask. Subsequently, the aqueous solution was heated to 110 °C (slight boiling) and kept for 5 h without stirring. After natural cooling to room temperature, the precipitate was centrifuged and washed with water several times, and the $Ni(OH)_2$ nanosheets were acquired.

*3.3. Synthesis of Pt/Ni(OH)₂ Nanosheets*

The prepared $Ni(OH)_2$ nanosheets were firstly dispersed in 1 mL of deionized water, and then 1 mL of PVP solution (90 mg/mL) was added. The added volume of the $K_2PtCl_4$ solution (200 mM) was based on the different Pt loadings (5, 10, 15, and 20 µmol denoted as $Pt_5/Ni(OH)_2$, $Pt_{10}/Ni(OH)_2$, $Pt_{15}/Ni(OH)_2$, $Pt_{20}/Ni(OH)_2$, respectively). After that, 20 µL of the HCOONa solution (100 mg/L) was dropped, and deionized water was utilized to control the total volume, fixed at 10 mL. After aging for 24 h, the precipitate was centrifuged and washed several times, and the $Pt/Ni(OH)_2$ nanosheets were acquired.

### 3.4. Characterizations

Scanning electron microscopy (SEM) was performed using a Hitachi S-3400N (Hitachi, Tokyo, Japan) with an accelerating voltage range from 0.3 to 30 kV. Transmission electron microscopy (TEM) and high-resolution transmission electron microscopy (HR-TEM) were performed using a Tecnai F20 (FETEM, 200 kV) (FEI, Hillsboro, OR, USA) and JEM-2100 (200 kV) (JEOL, Tokyo, Japan). Energy-dispersive X-ray spectroscopy (EDS) was carried out using an Oxford SDD X-ray (EDS) detector (Oxford Instruments, Abingdon, UK). The X-ray diffraction patterns of the as-prepared $Ni(OH)_2$ NSs and $Pt_n/Ni(OH)_2$ NSs ($n$ = 5, 10, 15, 20) were obtained on an X-ray Polycrystalline Diffractometer (D8 Advance, Bruker, Billerica, MA, USA) with a LYNXEYE detector and a Cu-K$\alpha$ radiation source ($\lambda$ = 1.5402 Å). The 2$\theta$ angle was scanned at a rate of 10° min$^{-1}$ from 10° to 80°. The Pt content of $Pt_n/Ni(OH)_2$ NSs was quantified by inductively coupled plasma-optical emission spectroscopy (ICP-OES) (Agilent 725 ICP-OES). The specific surface area of the as-prepared samples was obtained via the Brunauer–Emmett–Teller (BET) method using a Micromeritics ASAP 2460 with 4 ports. Fourier-transform infrared spectroscopy (FT-IR) was performed using a ThermoFisher iS20 (ThermoFisher, Waltham, MA, USA) paired with an iZ10 module. The X-ray photoelectron spectroscopy (XPS) measurements were collected by the Kratos Axis Supra$^+$ (Kratos Analytical, Manchester, UK) surface analysis instrument (energy resolution 0.45 eV) with Al K$\alpha$ radiation (1486.6 eV) under ultrahigh vacuum conditions. The catalytic reduction of 4-NP was evaluated via ultraviolet and visible spectrophotometry (UV-vis) using a Hach DR6000 UV-vis spectrophotometer (Hach, Loveland, CO, USA) with a scanning range from 300 nm to 500 nm and a scanning step of 1 nm.

### 3.5. Evaluation of Catalytic Performance

The catalytic performance was assessed using the room-temperature reduction of 4-NP to 4-AP in the presence of $NaBH_4$ as a model reaction. $Pt_n/Ni(OH)_2$ nanosheets as catalysts were firstly dispersed in deionized water at a concentration of 1 mg/mL. Typically, 4-NP (1 mL, 0.008M) and $NaBH_4$ (16 mL, 0.2M) were added to a 40 mL glass bottle, followed by the addition of 0.5 mL of $Pt_n/Ni(OH)_2$ aqueous dispersion, which was then slightly shaken to start the reaction. The sampling interval was set to 30 s, taking 1 mL of he reaction solution, filtrating it in a syringe filter, and diluting it by 2.5 times in a quartz cuvette with every sampling. The UV-vis absorption spectrum of the diluted reaction solution was measured to evaluate the catalytic performance. Before adding the $Pt_n/Ni(OH)_2$ aqueous dispersion, 1 mL of the initial mixture was sampled, diluted, and measured as a blank experiment. The conversion of 4-NP was evaluated after 10 min of reaction. The cyclic stability of $Pt_n/Ni(OH)_2$ was evaluated via 10 cycles of reaction.

## 4. Conclusions

In summary, we employed a two-step method to synthesize uniformly dispersed and loading-controllable Pt NPs, supported on $Ni(OH)_2$ NSs, as highly efficient heterogeneous catalysts. Notably, this method allows the Pt NPs to effectively anchor onto the pre-synthesized $Ni(OH)_2$ nanosheets at room temperature without high-temperature sintering. The structural characterizations confirmed the highly and uniformly dispersed Pt NPs with a size of 3 $\pm$ 0.66 nm were supported on the $Ni(OH)_2$ NSs with a high specific surface area. XPS analysis demonstrated that Pt anchoring on the $Ni(OH)_2$ nanosheets is strongly correlated with the oxygen vacancy of the $Ni(OH)_2$ substrate, in which the high oxygen vacancy density of nanostructured hydroxide provides sufficient anchor sites for Pt, leading to uniformly dispersed Pt NPs and controllable Pt loading. The $Pt_{10}/Ni(OH)_2$ NSs among all samples exhibited the most active and stable heterogeneous catalytic performance in the model reaction of the reduction of 4-NP to 4-AP in the presence of $NaBH_4$. Moreover, the catalytic activity of $Pt_n/Ni(OH)_2$ NSs surpassed the widely used commercial Pt/C catalyst, further affirming the advantages of our synthesized nanostructured $Ni(OH)_2$-supported small-size Pt NPs in heterogeneous catalysis. Our findings shed new light on

the advanced synthetic method of highly efficient noble-metal-supported catalysts for heterogeneous catalysis.

**Supplementary Materials:** The following supporting information can be downloaded at https://www.mdpi.com/article/10.3390/inorganics11060236/s1. Table S1: Mass loading of Pt on the synthesized $Pt_n/Ni(OH)_2$ NSs; Table S2: BET specific surface area of synthesized $Ni(OH)_2$ NSs and $Pt_n/Ni(OH)_2$ NSs; Table S3: Reaction rate constant $k$, mass normalized reaction rate constant $k_m$, and conversions of 4-NP at 10 min, catalyzed by $Pt_n/Ni(OH)_2$ NSs and commercial Pt/C; Table S4: Statistical analysis of conversion of 10 reaction cycles of $Pt_n/Ni(OH)_2$ NSs with different Pt loading; Figure S1: Magnified XRD patterns for as-prepared $Ni(OH)_2$ NSs and $Pt_n/Ni(OH)_2$ NSs with different Pt loading.; Figure S2: TEM images and particle size distributions of (**a**,**b**) $Pt_5/Ni(OH)_2$ NSs, (**c**,**d**) $Pt_{15}/Ni(OH)_2$ NSs, and (**e**,**f**) $Pt_{20}/Ni(OH)_2$ NSs.; Figure S3: TEM images of $Ni(OH)_2$ NSs.; Figure S4: Nitrogen adsorption–desorption isotherm of $Pt_{10}/Ni(OH)_2$ NSs.

**Author Contributions:** Experiments, J.-L.C.; data analysis J.-L.C. and Z.-L.L.; writing—original draft preparation, Z.-L.L.; writing—review and editing, J.-L.C., Z.-L.L. and H.-H.L.; supervision, H.-H.L. and C.-Z.L. All authors have read and agreed to the published version of the manuscript.

**Funding:** This research was supported by the National Natural Science Foundation of China (Grants 21838003, 21771170, 20080692), Shanghai Municipal Science and Technology Major Project, Shanghai Rising-Star Program (20QA1402700), Shanghai Sailing Program (20YF1410200).

**Data Availability Statement:** The data that support the findings of this study are available from the corresponding author upon reasonable request.

**Conflicts of Interest:** The authors declare no competing interest.

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
