# Peer review of "Pt Nanoparticles Supported on Ultrathin Ni(OH)2 Nanosheets for Highly Efficient Reduction of 4-Nitrophenol"

_inorganics, doi:10.3390/inorganics11060236_

Round 1

Reviewer 1 Report

This paper describes that Pt nanoparticle were deposited on Ni(OH)2 nanosheets and exhibited higher catalytic activity for 4-nitrophenol to 4-aminophenol in the presence of NaBH4 than the commercial Pt/C. The catalytic activity was correlated to the loading of Pt and Pt5/Ni(OH)2 and Pt10/Ni(OH)2 showed the highest activity while higher Pt loading decreased its activity, which is probably due to the increase in the Pt particle size. Although the catalytic activity of Pt5 and Pt10/Ni(OH)2 exceeds Pt/C, the characterization of the catalysts is required to revise according to the following comments.

1)     Although SEM and TEM images suggests the formation of nanosheet structure, formation of layered structure was not discussed on XRD patterns. The basal spacing can be estimated from the diffraction peak of the basal plane. If the Pt nanoparticles (NPs) are intercalated inside the layer, the basal spacing is supposed to expand.

2)     Pt particle size was discussed based on the specific surface area obtained by N2 adsorption (BET), and the authors stated that the increase in the surface area was correlated to the Pt particle size. Does it mean that Pt NPs are placed in the interlayer and that the expansion of the interlayer causes the increase in the surface area? None of explanation of the relation between the surface area and Pt particle size was given. If no, the particle size cannot be correlated to surface area because N2 is physisorbed on all surface including support. TEM or CO chemisorption (pulse CO) is required to estimate the Pt size. In addition, the relationships between Pt particle size (by TEM or CO chemisorption) and catalytic activity should be discussed.

3)     Although TEM image and size distribution of Pt10/Ni(OH)2 was given, other catalysts having different Pt loadings were not given.

4)     “n” of Ptn/Ni(OH)2 represents the amount of Pt added, but the actual loading (wt%) of Pt is easier to understand for readers.

5)     In introduction, the authors stated that Ni(OH)2 nanosheets have abundant vacancies, but none of references is cited. The authors stated that the oxygen vacancies worked as anchor sites. The explanation how the oxygen vacancies interact with Pt species should be described.

In p. 8, the catalytic activity order was connected with the BET surface area. However, the surface area cannot directly connect to the Pt particle size. The catalytic activity was correlated to the Pt particle size or the expansion of interlayer spacing, which enhances the diffusion of substrate to Pt inside the interlayer?

Reviewer 2 Report

Very interesting work, well justified and elaborated with the appropriate methodology. Interesting conclusions.

Just two formal recommendations: (1) Systematically insert blank space between last word and [citation]. I.e. last word[nn] must be typed ... last word [nn]. (2) in 2.1, ...platinate(II) should be said ...platinate(II).

Reviewer 3 Report

The manuscript titled "Pt nanoparticles supported on ultrathin Ni(OH)2 nanosheets for highly efficient reduction of 4-nitrophenol" it is evident that the authors have made decent contributions to the field of catalytic reduction of 4-nitrophenol using NaBH4 as a hydrogen source and Pt/Ni(OH)2 catalyst. In this study, Authors have doped different amounts of Pt into Ni(OH)2 nanosheets and thoroughly characterized their synthesis and analytical properties, which is impressive. However, it is important to note that the reduction of 4-nitrophenol using Pt-based heterogeneous catalysts is a well-known topic, and several hundred papers have been reported in the literature. Recently researchers have been more focused on non-noble metal-based heterogeneous materials; indeed, several non-noble metal-based materials have also been shown to exhibit excellent activity and selectivity for 4-NP at a wide range of temperatures (10.1039/d2ra02663e). Considering all there, it is strongly recommended that the authors extend the scope of their study to include the reduction of various -NO2 compounds, such as aromatic and aliphatic compounds. This would enhance the scope and strength of the manuscript and make it more advanced.

In their revised manuscript, the authors should consider presenting the following suggestions.

1. They should study and present graphs of H2 evolution from NaBH4 using Pt/Ni(OH)2 catalyst over time and various temperatures (25, 35, and 45/50 deg C) in the main manuscript. 

2. They should also calculate the rate of H2 evolution and its catalytic activity using the Arrhenius equation. 

3. They should present N2 adsorption and BET graphs in the manuscript/SI. This additional data would provide further insight into how porous supports play a role in catalytic activity and why non-porous materials are not as effective as porous-supported materials.

4. It is recommended that the authors compare their results with non-porous or bulky Ni nanoparticles in the literature to explain the impact of porous supports in catalysis. 

5. The authors should also consider citing the papers with the following references: (10.1039/d2ra02663e, 10.1039/d3ra01930f, 10.1021/acs.cgd.0c01028, and 10.1002/smll.201801233).

Round 2

Reviewer 2 Report

I recommend to accept the v2 version

Reviewer 3 Report

accept